# *teiresias*, a Fruitless target gene encoding an immunoglobulin-superfamily transmembrane protein, is required for neuronal feminization in *Drosophila*

Kosei Sato [1,3], Hiroki Ito[2,3] & Daisuke Yamamoto [1✉]

This study aims at identifying transcriptional targets of FruitlessBM (FruBM), which represents the major isoform of male-specific FruM transcription factors that induce neural sexual dimorphisms. A promoter of the axon-guidance factor gene *robo1* carries the 16-bp palindrome motif Pal1, to which FruM binds. Our genome-wide search for Pal1-homologous sequences yielded ~200 candidate genes. Among these, CG17716 potentially encodes a transmembrane protein with extracellular immunoglobulin (Ig)-like domains similar to Robo1. Indeed, FruBM overexpression reduced CG17716 mRNA and protein expression. In the *fru*-expressing mAL neuron cluster exhibiting sexual dimorphism, we found that CG17716 knockdown in female neurons completely transformed all neurites to the male-type. Conversely, CG17716 overexpression suppressed male-specific midline crossing of *fru*-expressing sensory axons. We renamed CG17716 *teiresias* (*tei*) based on this feminizing function. We hypothesize that Tei interacts with other Ig superfamily transmembrane proteins, including Robo1, to feminize the neurite patterns in females, whereas FruBM represses *tei* transcription in males.

[1] Neuro-Network Evolution Project, Advanced ICT Research Institute, National Institute of Information and Communications Technology, Kobe, Japan.
[2] Division of Neurogenetics, Tohoku University Graduate School of Life Sciences, Sendai, Japan. [3]These authors contributed equally: Kosei Sato, Hiroki Ito.
✉email: daichan@nict.go.jp

Reproductive success in many male animals relies on their behavioral performance during mating attempts with a female[1]. The neural circuitry that controls mating behavior has thus evolved under the strong pressure of sexual selection. Such an evolutionary drive led to the development of sexual dimorphisms in the neural circuitry and neurons that compose this circuitry[1]. *Drosophila melanogaster* is an outstanding model organism highly amenable to genetic dissection of complex traits, including mating behavior[2]. In this organism, the transcription factor gene *fruitless* (*fru*) plays a key role in organizing the sexually dimorphic circuitry for mating behavior by specifying sex-specific neuronal structures during development[3–5].

Among the four promoters identified in the *fru* gene, the most distal one (the P1 promoter) is dedicated to sex-specific functions ascribed to the male-specific translation of P1-derived mRNAs[6,7]. As a result of sex-specific splicing of the *fru*-P1 primary transcript, only male mRNAs have a long ORF encoding male-specific functional Fru proteins, called FruM. These proteins are translated as five isoforms, FruAM, FruBM, FruCM, FruDM, and FruEM, which share a common BTB domain in their N-terminus, but which each have a unique C-terminus[8]. The most prevalent isoform is FruBM, which has two zinc-finger motifs in the C-terminus, and forms a complex with chromatin regulators such as the TIF1 homolog Bonus (Bon), histone deacetylase 1 (HDAC1), and heterochromatin protein 1a (HP1a), to bind to more than 100 target sites on the genome for transcriptional regulation of downstream genes[9]. The best-characterized FruBM target is *robo1*[10], which encodes a transmembrane protein belonging to the immunoglobulin superfamily with an axon-guidance role in the developing nervous system[11]. FruBM binds to a 42-bp promoter segment of the *robo1* gene named the FruBM Response Obligatory Sequence (FROS), which contains a characteristic palindrome sequence of 16 nucleotides (Pal1)[10]. Partial deletion of Pal1 in the *robo1* gene impairs neural sexual differentiation and mating behavior in male flies, indicating that Pal1 is pivotal for FruBM in order to execute its sexual functions[10].

Many of the *fru*-expressing neurons are sexually dimorphic, including neurons composing the mAL cluster[12–14]. The mAL cluster displays sexual dimorphisms in the following four respects: (i) the number of neurons in the cluster (~5 in females vs. ~30 in males); (ii) the absence (females) or presence (males) of a neuronal subset carrying the ipsilateral neurite; (iii) the branched (females) vs. unbranched (males) tip structure of the descending contralateral neurite; and (iv) the focal (females) or expanded (males) terminal arbor distribution of the ascending contralateral neurite[15,16]. While these four types of sexual dimorphisms of mAL neurons are established in a *fru*-dependent manner, they are also regulated separately and independently from each other. For example, *Hunchback* knockdown in male mAL neurons feminizes only the descending contralateral neurite, whereas *robo1* knockdown feminizes only the ipsilateral neurite[17]. On the other hand, loss of the cell-death genes, *reaper*, *grim* and *head involution defective*, increases the number of cells composing the mAL cluster from ~5 to ~30 in females, but has no effect in males[16,18]. Therefore, FruBM likely interacts with different cofactors for transcription regulation, and/or FruBM acts on different targets to establish each of the four sexually dimorphic features of mAL neurons. With the aim of obtaining additional FruBM transcriptional targets for the establishment of neural sexual dimorphisms, we examined the effects of knocking down the genes harboring a Pal1-homologous motif on mAL neuron structures. We identified CG17716, a gene we call *teiresias* (*tei*), encoding another immunoglobulin-superfamily member with a transmembrane domain. Remarkably, knockdown of *tei* in female mAL neurons completely masculinized all three neurite characteristics, leaving the cell number unaffected. *tei* knockdown

also masculinized *fru*-expressing sensory neurons and mcALa interneurons in females. We propose that the Tei protein functions as a common receptor and interacts with a second receptor, e.g., *robo1*, which confers the ligand specificity on the heteromeric receptor complex, producing distinct neuronal populations that are diversified in their sex-specific structures.

## Results

**CG17716 is a Fruitless transcriptional target**. A genome-wide search for sequences similar to Pal1 in silico yielded ~200 sites, when replacements up to two nucleotides were allowed. Thirteen of these ~200 sites were located near the presumptive transcription start sites of genes implicated in neural development (Supplementary Data 1). We carried out real-time PCR analysis of these neural genes to identify possible changes in the transcript abundance in response to FruBM overexpression driven by *elav-GAL4* in neurons. We identified seven downregulated genes, i.e., CG17716 (*tei*), *Kr-h1*, *Tnks*, *robo3*, *Sema2a*, *Cip4*, and *ds*. Of these, CG17716 was particularly rich in Pal1-homologous motifs, with six Pal1-homologous sequences found within the 140 kb region harboring the gene (Fig. 1a–c).

CG17716 encodes a protein that contains five immunoglobulin-like repeats, a C-terminal transmembrane domain, and a signal peptide flanked by a putative cleavage site at the N-terminus (Fig. 1a). These structural features of the protein encoded by CG17716 are similar to those of Robo1 (Fig. 1d), which represents an established transcriptional target of FruBM[10]. Robo1 has been shown to play roles in the sexually dimorphic development of *fru*-expressing mAL neurons. Robo1 inhibits the formation of the ipsilateral neurite in females, while in males, FruBM disinhibits *robo1*, resulting in male-specific formation of the ipsilateral neurite. Robo1 has no role in the sex differences for cell number and branching patterns of the contralateral neurites of mAL neurons.

**Tei specifies the sex-type of brain interneurons**. The structural similarity of the CG17716-encoded protein and Robo1 prompted us to examine possible effects of CG17716 knockdown on the neuritogenesis in mAL neurons. We employed the MARCM technique[19] to clonally knock down CG17716 via the *fru*NP21 GAL4 driver in the mAL cluster (Fig. 2a–h). Reminiscent of the case of *robo1* knockdown, female mAL neurons with CG17716 knockdown developed an ipsilateral neurite (Fig. 2d arrows) that is otherwise absent in female neurons (Fig. 2c) in 40% of brain samples examined (Fig. 2i). Although the descending contralateral neurite of female mAL neurons with CG17716 knockdown had a bifurcating tip, the outer branch appeared much thinner than the inner branch (Fig. 2d arrowhead) in comparison with the counterpart in control female neurons (Fig. 2c). Quantification of fluorescent signals that monitor neuronal labeling along the imaginary transverse section drawn on the neurite terminals validated the observation that *tei* knockdown reduced the outer branch formation (Fig. 2j; see Supplementary Fig. 1 for the detailed method for quantifying fluorescent intensity of stained neurites). Our previous study using single-cell mAL clones with *fru* knockdown showed that the reduced outer branch in mAL neuroblast clones was a consequence of an increase in male-type neurons with the unbranched tip and a concomitant decrease in female-type neurons, which was an outcome of sexual transformation of a subset of mAL neurons[9]. Moreover, the ascending contralateral neurite in females with CG17716 knockdown acquired an extension that protruded toward the midline in the superior lateral protocerebrum of the brain (Fig. 2h asterisks) in the manner of the male neurite (Fig. 2e); in contrast, mAL neurons with *robo1* knockdown formed normal, round

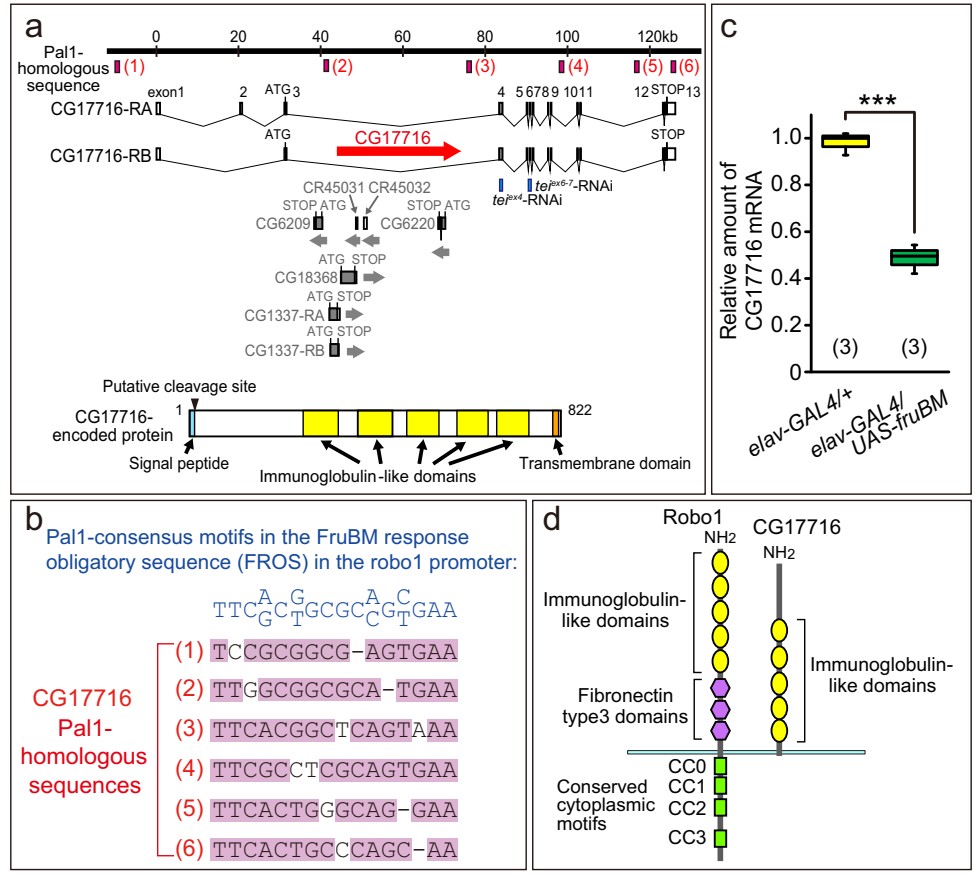

**Fig. 1 Identification of CG17716 as a candidate for the transcriptional target of FruBM. a** Genomic organization of CG17716: distribution of Pal1-homologous sequences in the CG17716 locus (upper panel) and structure of the CG17716-encoded protein (lower panel). **b** Six Pal1-homologous sequence stretches found in the CG17716 locus were aligned. The Pal1-consensus motif is shown above. **c** Reduction in the CG17716 mRNA amount by FruBM overexpression detected by real-time PCR. The box-and-whisker plot shows the first quartile (25th percentile), median, third quartile (75th percentile) and minimum and maximum of each set of data. Statistical differences were evaluated by the Student's $t$-test: \*\*\*$P < 0.001$. **d** Domain structures of the CG17716-encoded protein and Robo1. CC0–3: Conserved cytoplasmic motifs 0–3.

terminals typical of female neurons[10]. We conclude that CG17716 knockdown in female mAL neurons results in masculinization of the entire neurite structure. Because CG17716 functions as a feminizing factor in these cells, we renamed CG17716 *teiresias* (*tei*), after the blind prophet in Greek mythology who is transformed into a woman. The number of cells that comprise the mAL neurons was unaffected by knockdown of *tei* (Fig. 2k). Three independent *UAS-RNAi* fly lines, one from a stock center (line 102073) and two made in our laboratory (*UAS-tei^ex4-RNAi* and *UAS-tei^ex6-7-RNAi*), yielded qualitatively similar phenotypes; *UAS-tei^ex6-7-RNAi* gave the strongest effect, which reduced *tei* mRNA to ~6% of the control level (Fig. 2l).

We then asked whether *tei* plays a role in feminizing the neurite pattern in other sexually dimorphic *fru*-expressing neurons. Male mcALa neurons have small arbors near the cell body (Fig. 3a, e, arrow), which are absent in the female mcALa neurons (Fig. 3c, g). *tei* knockdown in female mcALa neurons induced the male-specific arbors (Fig. 3d, h, arrow), while *tei* knockdown in male mcALa showed no discernible effect on the neural structure (Fig. 3b, f, arrow). These observations were further verified by quantitative analysis that compared the distribution patterns of fluorescent signals around the protruding arbors labeled by GFP (Fig. 3i–k). In this analysis, the fluorescent intensity was measured from the base to the tip of the arbor shaft, and was transformed into a graphic representation, revealing that the intensity monotonically declined along the length of the shaft in male neurons (Fig. 3i). In females, in contrast, the analysis of

the corresponding field yielded a much steeper curve, which declined to the half-maximum at the distance of ~25% of the half-maximal distance in male neurons (Fig. 3j). *tei* knockdown doubled the half-maximal point in female neurons whereas it had no effect in male neurons (Fig. 3k). We conclude that *tei* is required for repressing the formation of the male-specific arbors in a female-specific manner. The cell number of the mcALa neurons was unaffected when *tei* was knocked down in either sex (Supplementary Fig. 2).

**Tei inhibits male-specific midline crossing of sensory axons**. Yet another example of *fru*-dependent sexual dimorphism is found in the axon projections of *ppk23*-expressing sensory neurons from the foreleg[20], which extend across the midline in the thoracic ganglia of males but not of females (Fig. 4a–d)[21–23]. We found that *tei* knockdown induces the midline crossing of *ppk23*-expressing axons in females (Fig. 4f vs. Fig. 4e, Supplementary Fig. 3). Quantification of the axon density by the midline crossing score (Fig. 4g, h)[22] revealed that *tei* knockdown significantly increased the trans-midline axons in females with no effect in males (Fig. 4i). In males, however, overexpression of *tei^+* significantly decreased midline crossing (Fig. 4j–l). In this series of experiment, *tei^+* was overexpressed with *EY11779*, a Gene-Search fly line with paired *UAS* sequences inserted near the *tei* locus on chromosome-2 (*UAS-tei^+*), so as to make chromosome 3 available for other transgenes to combine. Also, *Poxn-GAL4*, a

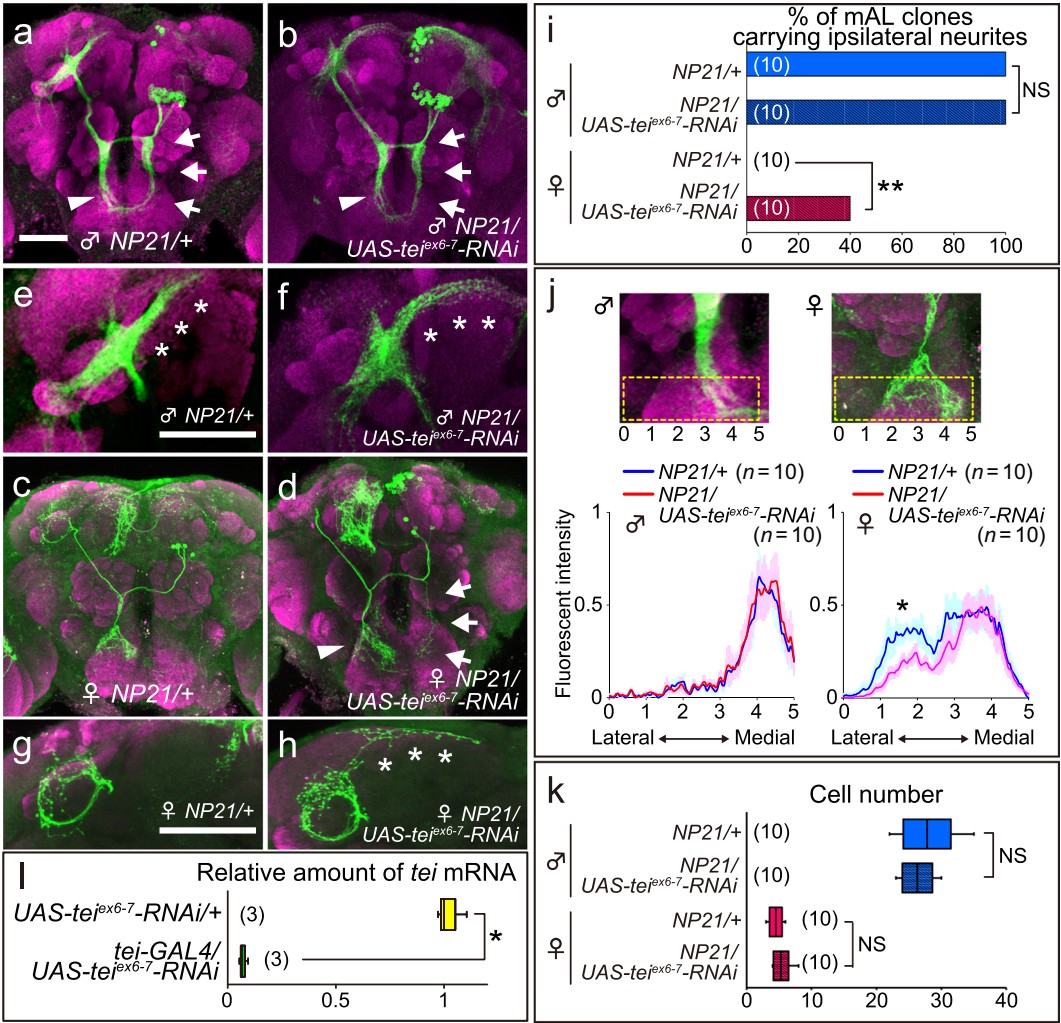

**Fig. 2 *tei* is involved in feminization of neurite structures of mAL neurons. a–d** mAL neuroblast clones without (**a**, **c**) or with (**b**, **d**) expression of *tei*[ex6-7] RNAi in female (**c**, **d**) and male (**a**, **b**) flies. The terminals of the ascending contralateral neurite are shown at a higher magnification in (**e–h**). Highlighted are the male-specific ipsilateral neurites (arrows), descending contralateral neurites (arrowheads) and male-type extensions of ascending contralateral neurites (asterisks). The intensity of fluorescence at every pixel along the horizontal broken lines drawn on the images was quantified to define the sex-type of the descending contralateral neurite (see below). Clonal neurons were visualized by mCD8::GFP (green) in brain tissues subjected to counter staining with the nc82 monoclonal antibody (magenta). The scale bar represents 50 μm. **i** Proportion of mAL neuroblast clones carrying the ipsilateral neurite without or with *tei*[ex6-7] RNAi expression. Statistical differences were evaluated by the Fisher's exact test: \*\**P* < 0.01. **j** Spatial distribution of fluorescence intensity along the imaginal transverse section of the brain images (regions encircled with broken lines in the upper panels were subjected to the analysis). The contralateral neurite extending to the suboesophageal ganglion was "optically sliced" at 6 levels, 0–5, and the fluorescent intensity measured along the horizontal line (lateral-medial axis) at each level is plotted at the bottom of the panel (here called an "arboplot") with the help of NIH ImageJ. An average trace of the 10 arboplots is illustrated (the mean ± SDM). More details of the method used for quantifying fluorescence intensity are given in Supplementary Fig. 1 and its legend. The tip of the descending contralateral neurite is bifurcating in females, resulting in two fluorescent peaks (right-hand side), whereas that of the male counterpart appears as tufts, resulting in a single prominent peak (left-hand side). **k** The number of cells within an mAL cluster without or with *tei*[ex6-7] RNAi expression. **l** Efficacy of knockdown of *tei* with *UAS-tei*[ex6-7] RNAi under the control of *tei-GAL4*. The number of confocal slices used for z-stacking was 62 (panel **a**), 70 (**b**), 100 (**c**), 85 (**d**), 32 (**e**), 29 (**f**), 47 (**g**), and 48 (**h**). The number of replicates is shown in parentheses. The box-and-whisker plot shows the first quartile (25th percentile), median, third quartile (75th percentile) and minimum and maximum of each set of data. The statistical significance of differences was evaluated by Mann–Whitney U-test (**k**, **l**): \**P* < 0.05.

pan-chemosensory driver, replaced *ppk23-GAL4* in another set of experiment, because the chromosome-3 carrying both a *fru* mutant allele and *Poxn-GAL4* was already available with no need to generate a new recombinant chromosome-3. We then asked whether *tei* knockdown in males would affect midline crossing in a *fru* mutant background (Fig. 4m–p). Notably, reduced midline crossing in *fru*²/*fru*^sat mutant males was alleviated by additional *tei* knockdown (Fig. 4n, p). Conversely, overexpression of *tei*⁺ further reduced midline crossing in *fru*²/*fru*^sat males (Fig. 4o, p). These observations are compatible with the hypothesis that *tei*

exerts a feminizer effect on the neurite pattern in different types of *fru*-expressing neurons, in which *tei* is negatively regulated by *fru*. FruBM is known to repress transcription of *robo1*, which otherwise inhibits the male-specific neurite formation[10]. In support of this notion, *robo1* knockdown via *ppk23-GAL4* promoted midline crossing in females (Fig. 4q, s). Importantly, additional expression of *tei*⁺ in these neurons counteracted the effect of *robo1* RNAi, resulting in a trace of midline crossing (Fig. 4r, s). Conversely, females with *tei* and *robo1* double knockdown had a midline crossing fiber bundle and this bundle was thicker than

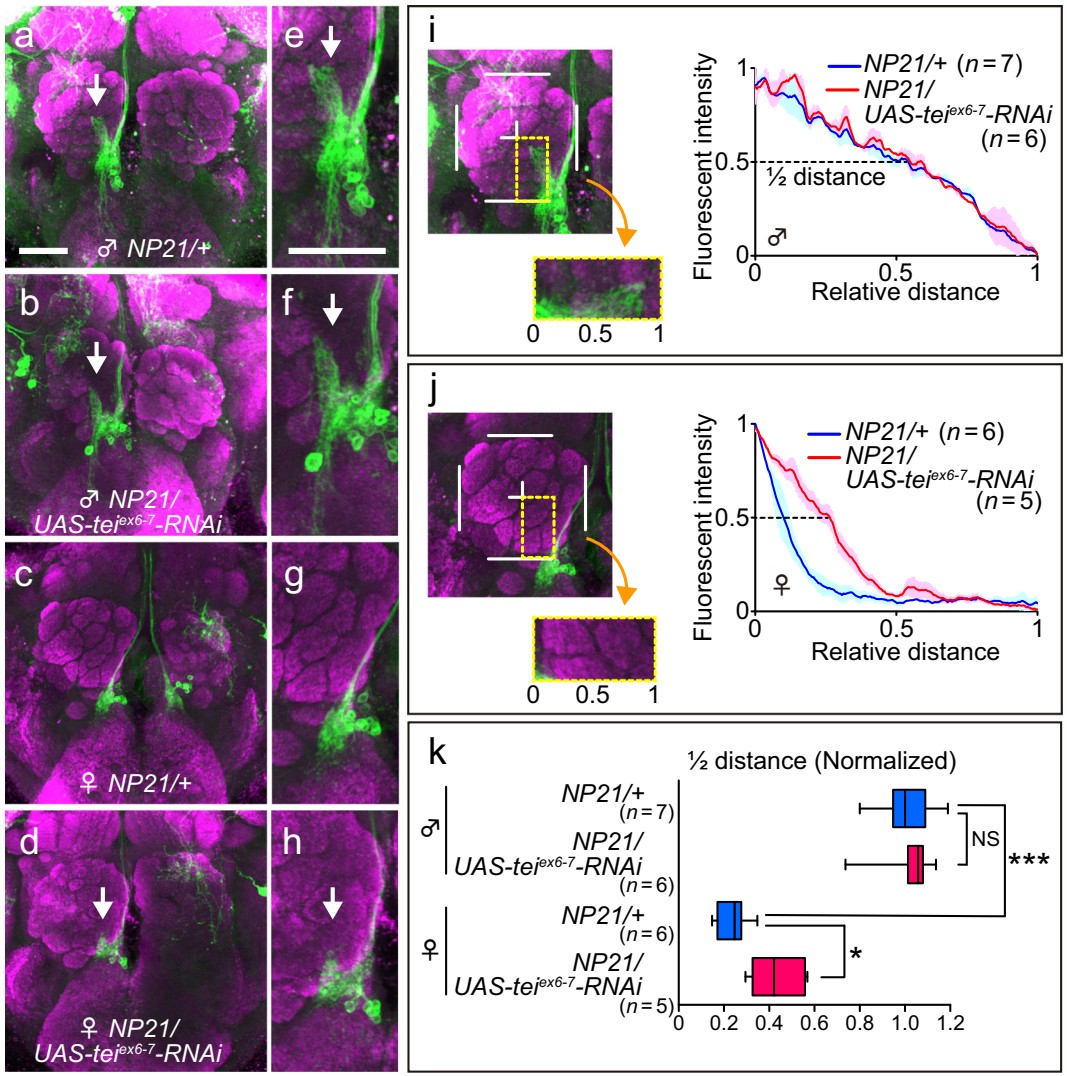

**Fig. 3 Effects of *tei* knockdown on the sexually dimorphic structures of mcALa neurons. a–d** mcALa neuroblast clones in a male (**a**) and female (**c**) control fly, a male (**b**) and female (**d**) fly with *tei* knockdown. The region that emanates male-specific arbors (arrows) is enlarged in (**e–h**). **i–k** Arboplot-aided analysis of the male-specific arbors. The region encircled with a broken line in the left-hand side panel was analyzed in males (**i**) and females (**j**) for the genotypes indicated. The half maximal distances as defined in arboplots in (**i**) and (**j**) were compared with and without *tei* knockdown in box plots shown in (**k**). Statistical differences were evaluated by the one-way ANOVA followed by Holm's multiple comparisons: ***$P < 0.001$, *$P < 0.05$; NS not significant. The scale bar represents 50 µm.

that formed in females with either *tei* or *robo1* single knockdown (Fig. 4t–w).

**Some *fru*-neurons may be refractory to FruM-induced *tei* repression.** To further validate the hypothesis that FruBM is a negative regulator of *tei* transcription, we generated an anti-Tei antibody and a *tei-GAL4* reporter line for visualization of potentially correlated changes in Tei and FruBM protein expression. The specificity of the anti-Tei antibody was confirmed by its ability to detect ectopic localization of Tei produced under the control of GAL4 drivers with established expression patterns (Supplementary Fig. 4). Furthermore, anti-Tei immunoreactivity of the wild-type CNS was markedly decreased in *tei* mutants (Supplementary Fig. 5). The *tei-GAL4* line was obtained by replacing the MiMIC cassette with the GAL4-encoding sequence at the Mi00744 insertion in the *tei* locus (Supplementary Fig. 5a–e). The reporter expression driven by *tei-GAL4* faithfully reproduced the anti-Tei staining pattern, validating its use for monitoring Tei expression (Supplementary Fig. 5f–k). *tei-GAL4*

labeled a large number of cells in the central nervous system (CNS) in both females and males, hindering the detection of any possible sex differences in its expression (Supplementary Fig. 5b–e). We therefore confined the reporter expression to a *fru*-positive subset of cells by the intersection of *tei-GAL4* and *fru^FLP^* and examined the CNS at the early pupal stage, when endogenous FruM expression commences to organize the sex-specific neural network (Fig. 5). At ~5 h after puparium formation (APF), a single cluster in the anterior lateral protocerebrum was found to express FruM strongly in males (Fig. 5d–f) but not females (Fig. 5a–c). This cluster expressed *tei-GAL4* in females (Fig. 5a, c) but not in males (Fig. 5d, f), exhibiting a sex difference. We, therefore, found an inverse correlation between Tei expression and FruM expression in this case. At ~12 h APF, most *fru*-expressing clusters, including the mAL cluster, were positive for FruM in the male (Fig. 5j–l) but not female (Fig. 5g–i) brain, in accordance with the male-specific nature of FruM. Notably, some of these FruM-positive cells in the male brain also expressed *tei-GAL4*, but others did not (Fig. 5j–l). In the male mAL cluster, for

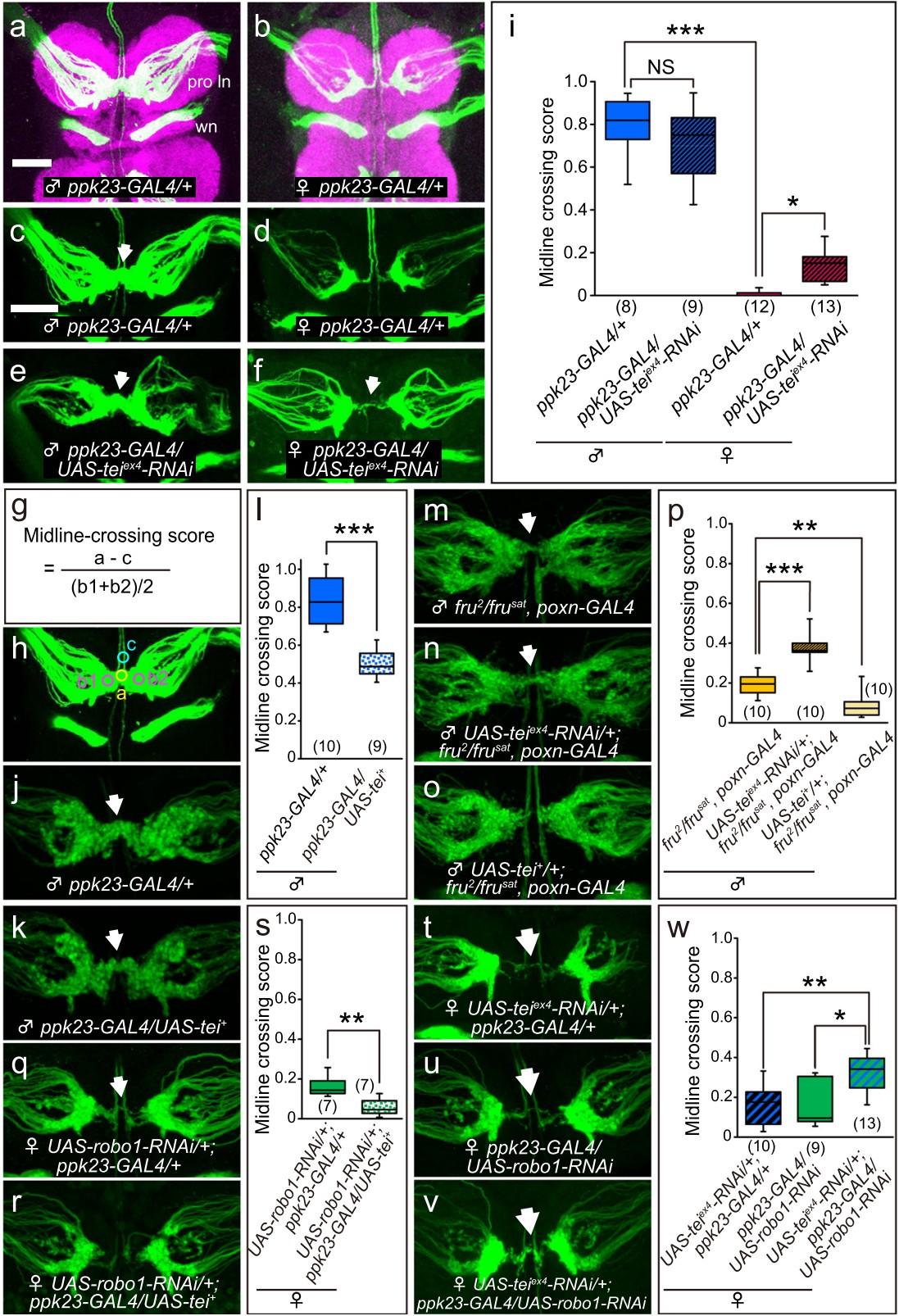

example (Fig. 5j–l), ~30 cells were labeled with the FruM antibody and only 4 of them expressed *tei-GAL4* at a detectable level (Fig. 5l). Interestingly, the intensity of Tei signals in FruM-expressing cells in males was as strong as that in females that lack FruM, despite a reduction in the number of Tei-positive cells in males. This observation might suggest that *tei* expression is repressed by FruM only in a subset of mAL neurons.

Alternatively, *tei* expression might be susceptible to FruM-mediated repression in a cell only within a narrow time window that does not temporally overlap across all cells, and thus some of the cells happened to be refractory to FruM-induced repression when the brain samples were prepared for examination. In the adult brain, most of FruM-expressing mAL neurons (~20 cells) were also positive for *tei-GAL4* presumably due to perdurance of

**Fig. 4 Effects of *tei* manipulations on the male-specific midline crossing of *ppk23*-expressing sensory axons. a–f** Central projections of *ppk23*-expressing axons in a male and female fly with (**e**, **f**) or without (**a–d**) *tei* knockdown. **g–i** The midline crossing score was calculated with the equation shown in (**g**) using the values of fluorescent intensity measured at sites a–c (**h**) and was compared in males and females with and without *tei* knockdown (**i**). Statistical differences were evaluated by the one-way ANOVA post hoc Turkey's multiple comparisons test: ***$P < 0.001$, *$P < 0.05$; NS, not significant. The scale bar represents 50 µm. pro ln, prothoracic leg neuromere; wn wing neuromere. **j–p** *fru*, *tei*, and *robo1* conjointly regulate midline crossing of sensory axons. **j–l** *tei*⁺ overexpression in males diminishes midline crossing. Examples of midline crossing (arrows) in males without (**j**) or with (**k**) *tei*⁺ overexpression and midline crossing scores of these two groups of males (**l**). **m–p** *fru* mutations reduce midline crossing (arrows) in males (**m**, **p**) and *tei* knockdown mitigates (**n**, **p**), while *tei* overexpression enhances (**o**, **p**), the *fru* mutant effect. **q–s** *robo1* knockdown in females promotes midline crossing (**q**, **s**) and *tei*⁺ overexpression counteracts this *robo1* knockdown effect (**r**, **s**). In this series of experiments, *tei*⁺ was overexpressed via EY11779. The number of replicates is shown in parentheses. **t–w** Enhancement of midline crossing in females with *tei* and *robo1* double knockdown (**v**, **w**) is stronger than that in females with *tei* (**t**, **w**) or *robo1* (**u**, **w**) single knockdown. The number of confocal slices used for z-stacking was 46 (panels **c**, **h**), 51 (**d**), 51 (**e**), 51 (**f**), 45 (**j**), 42 (**k**), 49 (**m**), 36 (**n**), 38 (**o**), 41 (**q**), 47 (**r**), 37 (**t**), 34 (**u**), and 37 (**v**). The box-and-whisker plots show the first quartile (25th percentile), median, third quartile (75th percentile) and minimum and maximum of each set of data. Statistical differences were evaluated by Mann–Whitney U-test (**l**, **s**) and Kruskal–Wallis analysis followed by Steel-Dwass multiple comparisons (**p**, **w**): ***$P < 0.001$, **$P < 0.01$, *$P < 0.05$. The scale bar represents 50 µm.

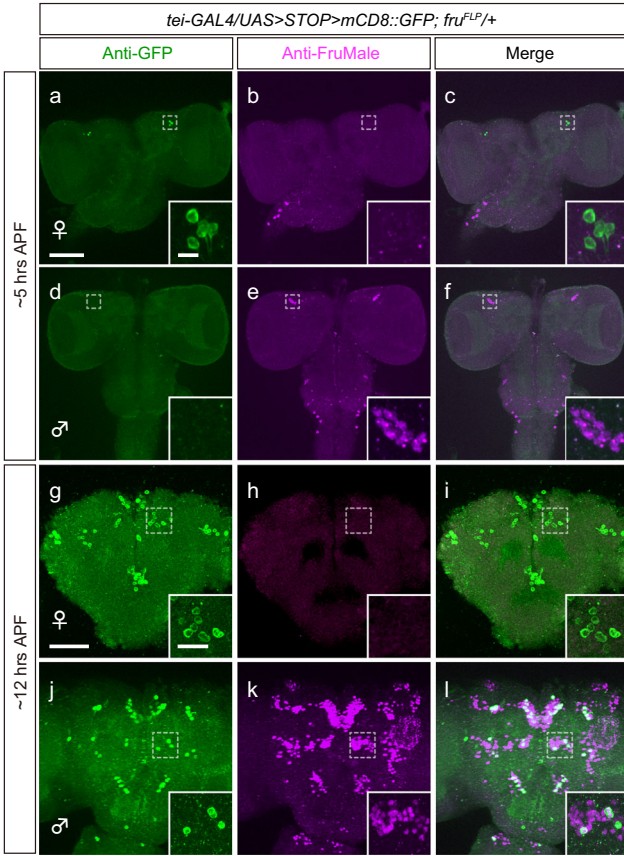

**Fig. 5 Sexually dimorphic Tei expression in *fru*-positive neurons was unraveled by the intersection of *tei-GAL4* and *fruᶠᴸᴾ*. a–f** At ~5 h APF, neurons within a single cluster in the anterior lateral protocerebrum (encircled with a broken-line frame and enlarged in insets) strongly express FruM (magenta) in males (**e**, **f**) but not females (**b**, **c**), and conversely, the same cells express *tei-GAL4* (green) in females (**a**, **c**) but not males (**d**, **f**). **g–l** At ~12 h APF, all known *fru*-positive clusters, including mAL (encircled with a broken-line frame and enlarged in insets), express FruM (magenta) in males (**k**, **l**) but not females (**h**, **i**), and *tei-GAL4* expression (green) is detected in not only females but also males, despite the fact that the number of *tei-GAL4*-positive cells is reduced in males. Scale bars represent 50 µm and 10 µm (for insets).

GAL4 protein (Supplementary Fig. 6), an observation that appears to favor the second possibility. We performed *tei* knockdown in *dsx*-expressing sexually dimorphic neurons in female flies while excluding the *fru*-positive subset of *dsx* neurons

via GAL80, and found no obvious change in the neuroanatomy thus visualized (Supplementary Fig. 7).

Next, we examined whether overexpression of FruM in the female CNS affects *tei* expression as monitored by *tei-GAL4* (Fig. 6a–f). We found a striking reduction in the intensity of anti-Tei staining upon FruM-overexpression in the third instar larval CNS (Fig. 6a–c vs. Fig. 6d–f), which otherwise does not express FruM and thus comparisons can be made without any interference from endogenous FruM expression. We conclude that FruM downregulates *tei* expression in a subset of *fru*-expressing neurons.

**Deficits in *tei* culminate in mating behavior defects.** To gain insights into the behavioral significance of the Tei-dependent sex-type specification of neurons, we examined the mating behavior of male flies, in which *tei*⁺ was overexpressed as driven by a *fru*-specific GAL4 to feminize *fru*-positive neurons in the wild-type or *fru* hypomorphic background (Fig. 6g). We found that the courtship activity was dramatically decreased in males upon *tei*⁺ overexpression in males irrespective of whether they were wild type or mutant for the *fru* locus (Fig. 6g). The reduction in courtship activities induced by *tei*⁺ overexpression was accompanied by a decrease in mating success (Fig. 6h). This result is compatible with the notion that *tei* has a feminizing activity on neurons that constitute sex-specific circuits for successful mating.

## Discussion

In this study, we identified *tei*, a new transcriptional target of FruBM, based on the in silico search for genes harboring the FruBM-binding consensus sequence and subsequent analysis of gene knockdown effects on sexually dimorphic neurite structures. Tei is a putative transmembrane protein carrying multiple immunoglobulin-like repeats in the extracellular portion. This structure is very much alike that of Robo1, the product of the best-characterized FruBM target gene. Indeed, *tei* knockdown in female mAL neurons induced the male-specific ipsilateral neurite as did *robo1* knockdown, revealing a functional similarity between *tei* and *robo1* in terms of the feminizing effect on this particular structure. Nonetheless, *tei* was distinctly different from *robo1* in that *tei* is also required for the female-typical shaping of ascending as well as descending contralateral neurites, in which *robo1* has no role. Robo1 has been shown to operate as a receptor for the secreted ligand Slit, activating several signal transduction pathways depending on the developmental context in both invertebrates and vertebrates[24]. Promiscuous interactions of Robo1 with other transmembrane receptors in *cis* have been implicated as a basis for the versatile performances of Robo1 in multiple developmental contexts. The lack of the cytoplasmic portion in Tei might suggest that Tei contributes to the ligand

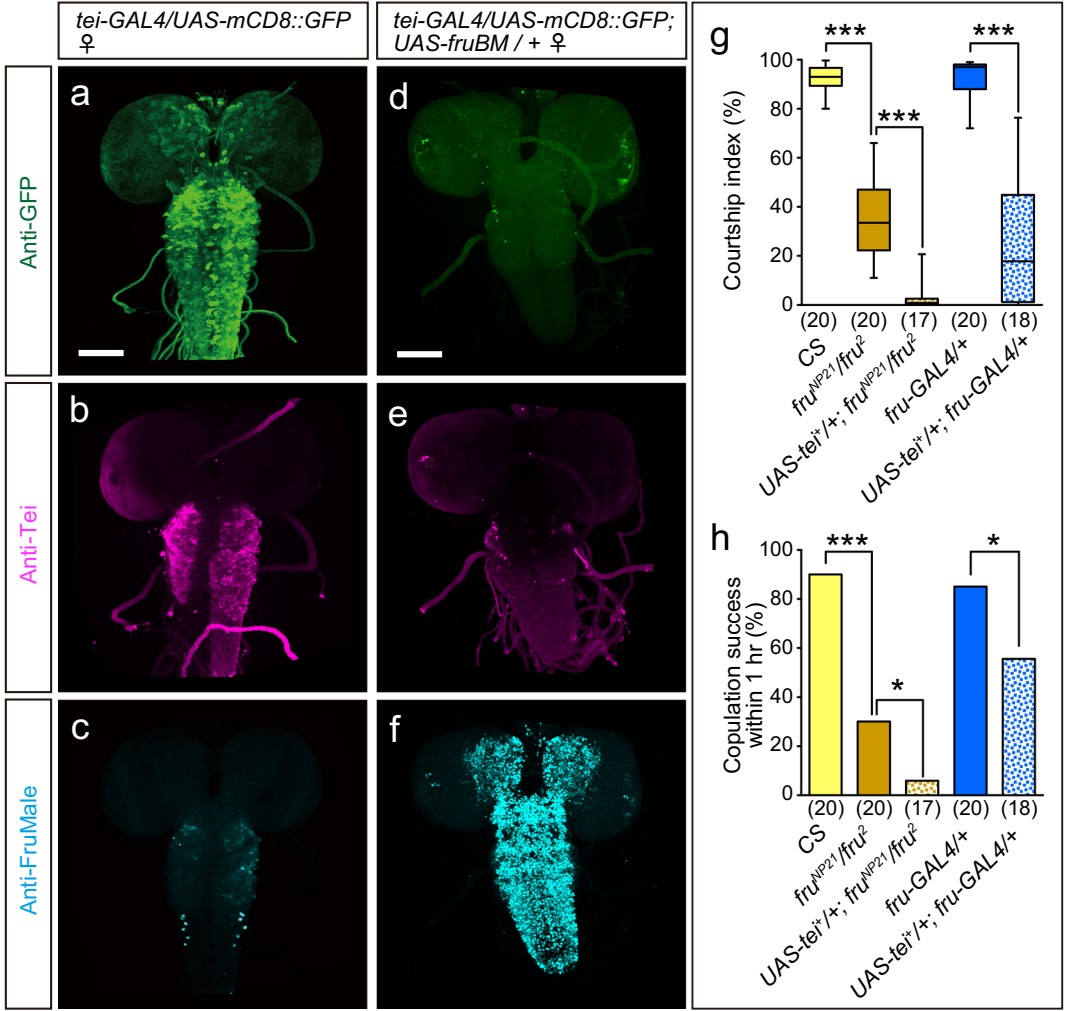

**Fig. 6 Overexpression of FruM represses *tei* expression in the larval CNS. a–c** The control CNS expresses a high level of Tei as detected by the GFP reporter for *tei-GAL4* (**a**) or anti-Tei antibody immunoreactivity (**b**) with little expression of FruM (**c**). **d–f** Overexpression of FruBM (**f**; detected by anti-FruM antibody) downregulates Tei expression as visualized by the reduced staining intensity for *tei-GAL4* expression (**d**) and anti-Tei antibody immunoreactivity (**e**). Scale bars represent 50 μm. **g, h** Courtship indices (**g**) and 1-h copulation success (**h**) of male flies with the genotypes indicated estimated when paired with a Canton-S wild-type female. Statistical differences were evaluated by the Kruskal–Wallis analysis followed by Steel-Dwass nonparametric multiple comparisons (**g**) and one-way ANOVA followed by Fisher's exact probability test (**h**). ***P < 0.001, *P < 0.05.

specificity whereas the specificity of intracellular signal transduction following heteromeric receptor activation is determined by Robo1 or another Tei partner that carries the C-terminal cytoplasmic domain. It is therefore plausible that Tei is one of these transmembrane receptors that cooperate with Robo1, and likely initiates together an intracellular biochemical cascade which ultimately inhibits the male-specific ipsilateral neurite from forming in an mAL neuron. Perhaps Tei associates with transmembrane receptors other than Robo1 in specifying the sex-specific structure in the ascending and descending contralateral neurites of an mAL neuron. Systematic knockdown in mAL neurons of genes encoding putative transmembrane receptors with immunoglobulin-like domains other than Robo protein members might lead to the identification of additional Tei partner proteins that are involved in the sex-type specification of the contralateral neurites.

Despite its striking effects on neurite structures, *tei* knockdown had no effect on the sexually dimorphism in the number of mAL neurons. This contrasts with the effects of *fru* loss-of-function mutation, which reduces the number of cells composing the mAL cluster in males from ~30 to ~5, the cell number typical of the mAL cluster in a wild-type female[16]. Removal of three major

cell-death genes, *hid*, *grim* and *rpr*, in female mAL neurons increased the number of cells composing the female mAL cluster up to 29 (the mean cell number was 19)[16]. This observation led to the proposition that cell death is the primary cause of the smaller number of cells composing the mAL cluster in females[16]. Moreover, the sex difference in the cell number of *fru*-expressing neurons has recently been attributed to a difference in proliferation in addition to a difference in cell death: male neuroblasts produce more cells than female neuroblasts, so that males have a larger number of cells even after blocking cell death[18]. These considerations lead to the supposition that FruM proteins separately control the neurite sex-type and the cell number sex-type. In this study we noted that the Pal1-homologous motif was not found in or around any of the *hid*, *grim*, and *rpr* loci, suggesting they are not direct transcriptional targets of FruM proteins. Alternatively, transcription of these genes might be regulated by a FruM isoform other than FruBM. In fact, the mAL cell number was shown to be affected by the *fru*[B2] (null for FruEM in our nomenclature) as well as the *fru*[C1] (null for FruBM in our nomenclature) mutation[25].

The 16-bp Pal1 core motif is located within the 42-bp FruBM-binding site, FROS. We first identified this motif in the *robo1*

gene by a combination of reporter assays, electromobility-shift assays, and CRISPR-Cas9-mediated in vivo mutagenesis in conjunction with phenotypic characterizations of single-cell clones of mAL neurons and mutant-fly behavior[10]. Using different strategies, two other research groups determined the binding consensus sequences for FruAM and FruEM in addition to FruBM. Neville et al.[26] in their study used the DNA adenosine methyltransferase identification (DamID) method, in which a Fru-fused bacterial methylase methylates the DNA around the genomic region to which the Fru moiety of the fusion protein binds. One of the reported FruBM-binding motifs in the DamID-based search revealed a 4/6 match when compared with Pal1, yet *robo1* was not obtained as a putative FruBM target in that study. Dalton et al.[27] used in vitro screening of oligo-DNAs with random sequences for binding to Fru zinc-finger motifs fused to glutathione S-transferase, revealing a binding consensus sequence for each isoform, but none of these had a similarity to Pal1. Of note, CG17716 (*tei*) has been included, together with another ~1400 genes, in a list of potential FruBM targets having a deduced FruBM-binding sequence by Neville et al.[26]. Reliable determination of consensus sequences for the binding of Fru isoforms and rigorous identification of Fru transcriptional targets are indispensable for obtaining a more complete picture of the molecular mechanisms underlying sex-specific circuit formation.

## Methods

**Fly strains**. Flies were reared on cornmeal-yeast medium at 25 °C. *fru^{NP21}*-*GAL4*[16], was used to label mAL neurons in MARCM. *5xUAS-HA-fruBM* (*UAS-fruBM*)[10] was used to overexpress FruBM. *elav-GAL4^{C155}* (*elav-GAL4*, #458), *10XUAS-IVS-mCD8::GFP* (#32186), *EY11779* (#20696), and other fly resources used in MARCM were obtained from the Bloomington *Drosophila* Stock Center. *UAS-CG17716 RNAi* (#102073) and *UAS-robo1 RNAi* (#42578) were obtained from the Vienna Drosophila Resource Center. *fru^{FLP}*, *UAS > STOP > mCD8::GFP*, and *p52a-GAL4* were gifts from B. Dickson. *ppk23-GAL4* was a gift from K. Scott.

**Behavioral assays**. For the analysis of mating behavior (Fig. 6g), males of each genotype were collected at eclosion and aged for 5–7 days. Each male fly was transferred to a small chamber (8 mm in diameter and 3 mm in height) with a Canton-S virgin female. The behavior of the fly pair was recorded using a video recorder. The courtship index (CI) was determined as the percentage of time that the male courted the females during a 5 min observation period. In calculating the CI, the time spent for all courtship elements, i.e., orientation, tapping, following, wing extension/vibration, and attempted copulation, was included. The copulation success (Fig. 6h) was calculated as the number of pairs that copulated during the observation period (60 min) divided by the number of total pairs observed.

**Generation of transgenic fly lines**. We made two *tei RNAi* lines, *UAS-tei RNAi^{ex6-7}*, and *UAS-teiRNA^{ex4}*, as well as *tei-GAL4* and *UAS-tei^+*. To obtain the *tei-GAL4* driver, we employed recombinase-mediated cassette exchange (RMCE)[28] at an MiMIC insertion within the *tei* locus. The *GAL4*-containing plasmid vector pBS-KS-attP2-SA(1)-T2A-GAL4-Hsp70 was injected together with ΦC31 plasmid DNA into the embryos of flies bearing the *Mi{Mic}00744* insertion in the *tei* locus. The recombinant transformant was signified by loss of the *yellow^+* selection marker, and the orientation of the *GAL4* insertion was determined by genomic PCR using the PCR primers Tei-F1 (5′-CGTGGATTATTGACCCGTTT-3′) and GAL4 R156 (5′-GCTTGGTCTTGGGGCTGTAG-3′).

*UAS-tei^+* transgenic lines were generated as follows. A full-length *tei* cDNA was attached by the sequence for the N-terminal HA tag, and cloned into the pUASTattB vector. The construct that generated was integrated into the attP2 site of the *D. melanogaster* genome using commercial microinjection services (Best-Gene Inc., Chino Hills, CA).

The WALIUM 20 vector (a gift from N. Perrimon) was used to make *UAS-RNAi* lines for *tei*. The 21nt sequences CCAGCATGTGGTACAAGAATG (for *UAS-tei^{ex4}* RNAi) and CGCACATCCGCAACCACTACC (for *UAS-tei^{ex6-7}* RNAi) were selected using the SnapDragon online tool (https://www.flyrnai.org/cgi-bin/RNAi_find_primers.pl). The corresponding oligos were synthesized, annealed, and ligated into the WALIUM 20 vector according to instructions available on the TRiP website (https://www.flyrnai.org/supplement/2ndGenProtocol.pdf). The plasmid vectors thus constructed were sequenced and injected into embryos that carried *attP2* as the landing site on the 3rd chromosome. Detailed information on primers can be found in Supplementary Table 1.

**Generation of anti-Tei antibodies**. A rabbit polyclonal anti-Tei antibody was raised against a 23-mer peptide, WPSISAEPADEVVDHRGGGKPAK, which was encoded by exon 4 of the *tei* gene (residues 56–78 of Tei) (GenBank accession number NP_523731), and was affinity purified.

**Dissection, immunohistochemistry and imaging of the central nervous system (CNS)**. For immunostaining of the adult brain, the brains of 2- to 7-day-old flies of the relevant genotypes were dissected in PBS and fixed in 4% paraformaldehyde in PBS for 60 min on ice. After washing in PBT (PBS with 0.5% Triton X-100), the tissues were kept in PBTN (PBT with 10% normal goat serum) for 60 min at room temperature (rt) for blocking, and reacted in PBTN for 24 h at 4 °C with the following primary antibodies and dilutions: rabbit anti-GFP at 1:500 (Invitrogen, A6455); mouse nc82 at 1:10 (Developmental Studies Hybridoma Bank); guinea pig anti-FruMale at 1:500[29]; chicken anti-GFP at 1:500 (abcam, ab13970); rabbit anti-Tei at 1:100 (this study); and rat anti-HA at 1/500 (Roche, 11867423001). After 1 h of washing in PBT, the tissues were incubated for 24 h at 4 °C in the following secondary antibodies and dilutions: Alexa Fluor488 anti-rabbit IgG; Alexa Fluor546 anti-mouse IgG; Alexa Fluor546 anti-guinea pig IgG; and Alexa Fluor546 anti-rat IgG (all at 1/200 and all from Invitrogen). Samples were washed for 1 h in PBT before being mounted with VECTASHIELD mounting medium (Vector Laboratories Inc.). Stacks of optical sections at 1 μm were obtained with a Zeiss LSM 510 META confocal microscope and processed with the software packages NIH Image J (http://rsb.info.nih.gov/ij/) and Adobe Photoshop.

**Clonal analysis of mAL and mcALa neurons**. To generate neuroblast clones of mAL (Fig. 2a–h) and mcALa (Fig. 3a–h) neurons, we employed the MARCM method. *fru^{NP21}-GAL4* was used as a GAL4 driver and embryos were heat shocked (37 °C) at 0 to 24 h after egg-laying in a water bath for 1 h.

**Quantification of fluorescent intensity of stained neurites**. The quantification of the fluorescent intensities of mcALa neurons (Fig. 3i, j) was carried out as follows. First, size variations across individual brains were normalized by using the unambiguous landmark structure antennal lobe, with which mcALa somata are juxtaposed. The dorsal (anterior), ventral (posterior), and two lateral (left and right) limits of the anterior lobe contour were delineated as shown in the two sets of perpendicular lines on the illustrated z-projection images of the brain. These two sets of lines were used to define the center of the antennal lobe. The center of the antennal lobe was then used to position an imaginal rectangle so that one of its upper corners (the inner corner) was superimposed on the antennal lobe center while the other upper corner (the outer corner) was placed at the midpoint between the antennal lobe center and the line that delimits the medial end of the antennal lobe (indicated by the box drawn with a broken line in Fig. 3i, j, left-hand side panels). The height of the rectangle is plotted along the $x$ axis starting from the point closest to the base (i.e., point 0 on the surface of a soma) to the top of the rectangle. The fluorescent intensity values at the same height were summed within the rectangle and plotted on the $y$ axis as shown in the right-hand side panels of Fig. 3i, j. The intensity curves thus obtained represent the spatial distribution of anti-body labeled-arbors, which are sexually dimorphic for mcALa neurons. The details of quantification of stained neurites of mAL are shown in Supplementary Fig. 1 and described in the corresponding legend.

**Midline crossing score analysis**. The Midline crossing score (MCS) was calculated as described previously[9]. Briefly, stacked images of each sample were summed up using ImageJ. Then, a circle of 8 μm was drawn on the resultant image so that the circle was centered at the prothoracic midline, where trans-midline axons are expected to run in males. A few sensory neurons originating from prothoracic legs send ascending axons directly to the brain, which turn perpendicularly within the prothoracic ganglion. The midpoint of bilateral ascending fibers at their turning points was used to center the circle. The fluorescent intensity within the circle marked "a" was measured to quantify the level of midline crossing by fibers. Similarly, the fiber tracts located lateral to the midline were quantified within circles "b1" and "b2" for fluorescent intensity. To normalize for the background fluorescent level, the areas with no fibers delineated by circles "c" were also measured. Finally, the MCS was calculated as: $MCS = (a - c)/[(b1 + b2)/2]$.

**Real-time PCR**. Real-time PCR was performed using a LightCycler 1.0 system (Roche). Total RNA was extracted from the brains of 3rd instar female larvae with FruBM-overexpression using an RNeasy Mini Kit (Qiagen, 74104). To quantify the expression levels of transcripts from genes harboring a Pal1-homologous sequence, equal amounts of RNA were used to synthesize cDNA using a ReverTra Ace qPCR RT kit (TOYOBO, FSQ-101). Each cDNA was mixed with SYBR Premix Ex Taq II (TAKARA, RR820S) and 5 pmol of both forward and reverse primers. The forward and reverse primers for *tei* were 5′-TCGATAAGGCGTGAACATTCCGG-3′ and 5′-CTGCCACGGAAACATCCTGATCA -3′, respectively. *RpL32* (*rp49*) was amplified as an internal control using the primer pair 5′-AGATCGTGAAGAA GCGCACCAAG-3′ (forward) and 5′-CACCAGGAACTTCTTGAATCCGG-3′ (reverse). Real-time PCR was conducted at 95 °C for 30 s (initial denaturation), followed by 40 cycles of denaturation at 95 °C for 5 s, annealing at 55 °C for 30 s

and elongation at 72 °C for 30 s. Data processing was performed using LightCycler Software Ver. 3.5 (Roche).

**Database searches for Pal1-homologous sequences**. The *D. melanogaster* genome BDGP Release 6 + ISO1 MT/dm6 (August 2014) was searched for Pal1-homologous sequences with the aid of the GGGenome (http://gggenome.dbcls.jp/en/) search engine, allowing for a maximum of two mismatches.

**Protein motif search for the CG17716-encoded protein**. To search for protein motifs in the CG17716-encoded protein, InterPro (https://www.ebi.ac.uk/interpro/), the SignalP 4.1 Server (http://www.cbs.dtu.dk/services/SignalP/), the TMHMM Server v. 2.0 (http://www.cbs.dtu.dk/services/TMHMM/), Phobius (http://phobius.sbc.su.se/), and the "DAS"-Transmembrane Prediction Server (https://tmdas.bioinfo.se/DAS/index.html) were used.

**Statistics and reproducibility**. Statistical analyses were done by GraphPad Prism 7.0b software. Experiments were repeated independently on multiple flies as mentioned in the text and figure legends.

**Reporting summary**. Further information on research design is available in the Nature Research Reporting Summary linked to this article.

## Data availability

The data sets generated for this paper are available from the corresponding author upon reasonable request.

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

## Acknowledgements

We thank Barry Dickson, Kristin Scott, the Bloomington *Drosophila* Stock Center and the Vienna Drosophila Resource Center for fly stocks and Y. Takamura for secretarial assistance. This work was supported in part by Grants-in-Aid for Scientific Research from the Ministry of Education, Culture, Sports, Science and Technology (16H06371, 19H04923, to D.Y.; 17K07040, 19H04766 to K.S.), a Life Science Grant from the Takeda Science Foundation to K.S., and a Hyogo Science and Technology Association Grant to K.S.

## Author contributions

Conceptualization: H.I.; methodology: H.I.; investigation: K.S., H.I.; writing—original draft: D.Y.; writing—review and editing: K.S., D.Y., H.I.; visualization: H.I., K.S.; supervision: D.Y.; project administration: D.Y.; funding acquisition: K.S., D.Y.

## Competing interests

The authors declare no competing interests.
