## [Peer Review File · Communications Biology]

Reviewers' comments:

Reviewer #1 (Remarks to the Author):

This manuscript by Sato et al., identifies a new Fruitless target gene, *teiresias*, that functions to feminize neuron structure in the *Drosophila* adult nervous system. Nervous system sexual dimorphism is found in many organisms and the transcriptional control of sex-specificity in terms of neuron function and anatomy is of considerably interest to many in the developmental and behavioral neuroscience community. This paper is very well written with an informative introduction and clear hypotheses and outcomes. Work from the Yamamoto lab is consistently thorough and the data (using appropriate statistical methods) supports their conclusions. However, presentation of the data in a few figures and the text needs clarification.

Major points:

1. Perhaps I missed this but somewhere, the figure legend, the text or a table, the entire genotype for each experiment/panel must be given.
2. More information is needed in regards to *tei*-Gal4 images, are the adult brains z-projections or specific optical sections. Some of the cell bodies in the adult brain images are very large (SFig. 4) – does this indicate anterior or posterior sections?
3. Based on the images in SFig. 4, anti-*Tei* expression does not look to be found on axons whereas *Robo* is found on commissures, the authors should address if this is a matter of antibody strength or other considerations.
4. The authors should include copulation success in Fig. 7 even with a low courtship index, the key is whether this reduces copulation. It would be best to redo the experiment using *tsh*-Gal80 especially as there looks to be widespread VNC expression (in the larva at least). This would be possible as *tsh*-Gal80 is on the second chromosome.

Minor points:

1. The arbors in Fig. 2c are obscured by NP21
2. In the text, a description of Fig. a' and c' are missing.
3. Fig. 2F is difficult to understand, what does the dashed box represent? The fluorescent intensity experiment in Fig. 3 is better described.

Reviewer #2 (Remarks to the Author):

How the male-specific Fruitless transcription factors specify the development of at least some sex-specific neurons in *Drosophila* is key to understand how a master gene like *fru* controls sexual dimorphic development and behaviors. Studies from the Yamamoto lab have identified some key factors that function with *Fru* or downstream of *Fru*, and in this manuscript, they continue to identify *Fru* target genes and found CG17716, or *teiresias* as they named by its function. They found that *tei* is down regulated by an isoform of *FruM* (*FruBM*), and is required for neuronal feminization in *Drosophila* females. They demonstrated *tei*'s feminization role in three different subsets of neurons. They also showed that *Tei*'s expression is sexually dimorphic at least in certain *fru* neurons. Overall, this is an important study adding to the field, though the detailed mechanism of how *Tei* functions to feminize neurons is not elucidated.

I only have two minor points for this manuscript:

1. I would suggest combining Fig 4 and Fig 5 as they are related data on *Tei*'s role in midline-crossing of gustatory neurons. Since they showed that knocking down *Tei* or *Robo1* increased

midline crossing in females, it is therefore important to see if simultaneously knocking down Tei and Robo1 would have a stronger effect on midline crossing. This at least could test whether Tei and Robo1 function together or in parallel.

2. I am wondering why the authors keep changing the developmental stages of flies that were used for immunostaining, some with adult, and some with larvae or pupae. I do not have problem with using either developmental stage for certain experimental purpose, e.g., whether over-expression of FruBM would inhibit Tei expression in larvae or other stage, but the authors are at least should mentioned the logic between using flies with different developmental stages for different experiments.

Brief summary of the manuscript

Understanding the molecular mechanism of how behavioural neural circuits are developed is the holy grail of neuroscience. The Fruitless protein (Fru), which functions to establish sexually dimorphic courtship behavior in flies, provides an excellent model to unravel this puzzle. Fru is a Zn-finger containing transcription factor with other structural domains that may form protein-protein interactions with other protein partners. FruBM is the male-specific isoform of Fru. This manuscript describes the identification of the second FruBM gene target known to date, Teiresias (Tei). Robo1 is another FruBM target, also discovered by the same group.

The authors identified Tei as a possible gene target of FruBM through a genome-wide bioinformatic analysis of promoter sequences that contain the previously established Pal1 sequence as the essential binding site for FruBM. The authors proved that Tei is a target of FruBM by using different genetic manipulations to overexpress and knockdown Tei, FruBM, Robo, and then evaluating the phenotypic consequences of the morphology of several well-characterized sexually dimorphic regions of the Fru circuit. The authors concluded that 1) FruBM represses the expression level of Tei, both at the RNA and protein level; and 2) Tei has a feminizing function, responsible in developing feminized neurite patterns when it is expressed; in contrast, Tei has no effect on the cell number in clusters that show sexually dimorphic neuronal population. Finally, the authors hypothesized that the mechanism of Tei function may be to interact with other IgG domain containing proteins, like Robo1, which may lead to a change in intracellular signaling that alters neurite morphology.

Overall impression of the work

This manuscript contains a body of meticulously designed experiments that establishes Tei as the second gene target of FruBM. The authors presented a thought-provoking hypothesis that Tei may exert its function by forming a heteromeric complex with Robo, though the data presented may suggest otherwise – see my comments below. The hypothesis is interesting as the use of the immunoglobulin-like structural domain to generate functional variability is a recurrent theme observed in other fields. I think the authors can develop this hypothesis further by discussing more about how Slit/Robo1 signaling works. For example, can Robo1 work on its own? What is the stoichiometry of a functional Robo1 receptor? Is there anything known about what the C-terminal domain of Robo1 does during Slit/Robo1 signaling? These questions are relevant to Tei as it does not have a C-terminal domain like Robo, according to Fig 1a. If Tei were to interact with Robo, how will this affect Slit/Robo1 signaling?

Specific comments, with recommendations for addressing each comment

P7 line 101: What are the 7 genes (assuming CG17716 is one of the seven) that are downregulated? The cited Supplementary Table 1 does not seem to be the right table intended for this sentence.

P11 lines 167-168: You may want to emphasize that there are two tei RNAi lines that confirm this observation here, since you're citing results from both lines.

P11-12 lines 170-182 and Figures 5d-g: Why did you use a different Gal4 line for the overexpression experiment? I assume poxn-Gal4 is EY11779? Why not use PPk23-Gal4 with *fru²/fru^{sat}*? I assume for practical reasons, but some explanation is needed.

P12 lines 180-181: "... *additional* ... *tei*⁺ in these neurons *counteracted* ... to *increase* midline crossing.." but you're actually looking for a decreased midline crossing in the cited Figure. I understand the meaning, but the sentence is a bit convoluted. Rewording or rephrasing may improve readability.

P12 lines 178-182, P16 lines 252-260

FruBM represses both Robo1 and Tei activity. Both Robo1 and Tei KD promote midline crossing. Results show that overexpression of Tei can counteract Robo1 KD effect. Wouldn't this observation disprove the hypothesis stated in the abstract (p2 lines 29-30) that "Tei interacts with other IgG superfamily transmembrane proteins, including Robo1, to feminize the neurite patterns in females..."? If Tei/Robo1 functions as a heterodimer, then KD of one gene should not be rescued by the OE of another? Can these receptors function as homodimers? Tei lacks a substantial C-terminal domain (Fig 1d). What does it mean when it interacts with other IgG transmembrane proteins with a larger C-terminal domain? Will it inhibit the intracellular effect completely or just attenuate it?!?

P14 lines 210-219: The observation that the intensity of Tei signals in FruM-expressing cells is strong is very interesting. The authors offer two explanations: 1) FruM only represses Tei expression in a subset of mAL neurons; and 2) There's a temporal lag between FruM expression and Tei expression. Can't possibility 2 be tested by fine-tuning the developmental time points for the co-staining analysis in the experiments for Fig 6? If possibility 1 is true, then there must be an unidentified factor that stops FruM from repressing Tei expression in a subset of mAL neurons? Could this factor be Fru/Bon/HP1a?!? For example, FruM is present but it's in complex with the unknown factor that prevents it from binding to the promoter of Tei?

P14 lines 216-219: The Dsx experiment doesn't seem to belong in this section. Instead, the first paragraph of the next section (p15 lines 222-226) should go after line 216 on p14?

P20 Fly strains: explain somewhere what poxn-Gal4 is.

P23 line 348: missing a bracket

P24 line 377: "so that the circle was centered ..." how do you ensure this is done consistently from sample to sample?

P35 lines 534-537: More description of the arboplot would be nice. For example, is there any consistent landmark you use to draw the yellow dotted box for optical slicing for each sample? Does it need a special plugin in Image J? Maybe useful to write a method section to describe how this is done. It's a very nice quantification procedure that other researchers will find useful to implement.

P37 line 576: EY11779/Poxn-Gal4?

P38 Figure 6: How do you know the single cluster that expresses Tei in female is the same cluster that expresses Fru in male (Figure a-f), since their expression is supposedly mutually exclusive?

P35 line 544, P40 line 611:

“The box plot shows the median and 10th, 25th, 75th, 90th percentiles”? Please correct.

Point-by-point responses to the reviewers' comments

Reviewer #1

Major points:

Q1. Perhaps I missed this but somewhere, the figure legend, the text or a table, the entire genotype for each experiment/panel must be given.

A1. The entire genotype for each experiment is given in Supplementary Table 3 of the revised manuscript.

Q2. More information is needed in regards to *tei*-Gal4 images, are the adult brains z-projections or specific optical sections. Some of the cell bodies in the adult brain images are very large (SFig. 4) – does this indicate anterior or posterior sections?

A2. The brain images shown are z-projections. This fact is described in the legend to Supplementary Fig. 5 in the revised manuscript (line 70). Please note that the current Supplementary Fig. 5 corresponds to the previous Supplementary Fig. 4.

Q3. Based on the images in SFig. 4, anti-*Tei* expression does not look to be found on axons whereas *Robo* is found on commissures, the authors should address if this is a matter of antibody strength or other considerations.

A3. In the images shown in Supplementary Fig. 5 (previous Supplementary Fig. 4), the cell bodies were in focus, because coexpression of the anti-*Tei* antigen and the GFP reporter was most obvious in this portion of the neurons. The anti-*Tei* antibody stained fibers as well, as can be seen in the image newly added to Supplementary Fig. 5 (Panel j of the new Supplementary Fig. 5 shows a magnified view of the neurites of a few optic lobe neurons from a CS female stained with the anti-*Tei* antibody).

Q4. The authors should include copulation success in Fig. 7 even with a low courtship index, the key is whether this reduces copulation. It would be best to redo the experiment using *tsh*-Gal80 especially as there looks to be widespread VNC expression (in the larva at least). This would be possible as *tsh*-Gal80 is on the second chromosome.

A4. We reconducted our mating behavior assays to quantify the effect of *fru*[+] overexpression in males on mating success over a 1-hour-observation period. The reduction in courtship activities induced by *tei*[+] overexpression was accompanied by a decrease in mating success, as described in lines 247–249 and illustrated in Fig. 6h of the revised manuscript (please note that the current Fig. 6 corresponds to the previous Fig. 7).

Minor points:

Q5. The arbors in Fig. 2c are obscured by NP21

A5. We adjusted the label “NP21” in Fig. 2c so that it no longer obscures the neural arbors.

Q6. In the text, a description of Fig. a and c are missing.

A6. We confirmed that all figures were referred to in the main text.

Q7. Fig. 2F is difficult to understand, what does the dashed box represent? The fluorescent intensity experiment in Fig. 3 is better described.

A7. Figure 2F sums up the intensity values of all pixels along the same meridian points (but different parallel points) for the region contoured by the dashed line (upper panels) and plotted on a coordinate plane (lower panels). The details of the method used are described in the legend of Supplementary Fig. 1, which is dedicated to an explanation of the quantification of fluorescent intensities of stained neurites of mAL neurons. The quantification of the fluorescent intensities of mcALa neurons (Fig. 3e, f) was similarly carried out with some modifications as follows. First, size variations across individual brains were normalized by using the unambiguous landmark structure antennal lobe, with which mcALa somata are juxtaposed. The dorsal (anterior), ventral (posterior) and two lateral (left and right) limits of the anterior lobe contour were delineated as shown in the two sets of perpendicular lines on the illustrated z-projection images of the brain. These two sets of lines were used to define the center of the antennal lobe. The center of the antennal lobe was then used to position an imaginal rectangle so that one of its upper corners (the inner corner) was superimposed on the antennal lobe center while the other upper corner (the outer corner) was placed at the midpoint between the antennal lobe center and the line that delimits the medial end of the antennal lobe (indicated by the box drawn with a broken line in Fig. 3e, f, left-hand side panels). The height of the rectangle is plotted along the x axis starting from the point closest to the base (i.e., point 0 on the surface of a soma) to the top of the rectangle. The fluorescent intensity values at the same height were summed within the rectangle and plotted on the y axis as shown in the right-hand side panels of Fig. 3e, f. The intensity curves thus obtained represent the spatial distribution of anti-body labeled-arbors, which are sexually dimorphic for mcALa neurons. The above explanation of the procedure used to obtain a quantitative presentation of the fluorescence distribution on a neuronal image is given in the Methods section of the revised manuscript (lines 396–415).

Reviewer #2 (Remarks to the Author):

I only have two minor points for this manuscript:

Q8. I would suggest combining Fig 4 and Fig 5 as they are related data on Tei's role in midline-crossing of gustatory neurons. Since they showed that knocking down Tei or Robo1 increased midline crossing in females, it is therefore important to see if simultaneously knocking down Tei and Robo1 would have a stronger effect on midline crossing. This at least could test whether Tei and Robo1 function together or in parallel.

A8. We conducted an additional experiment to evaluate the level of midline crossing in flies with *tei* and *robo1* double knockdown according to the suggestion by the reviewer. As shown in panels r–u of revised Fig. 4 (which integrates the former Figs. 4 and 5), females with *tei* and *robo1* double knockdown had a midline crossing fiber bundle and this bundle was thicker than that formed in females with either *tei* or *robo1* single knockdown (lines 190–192). The result was clear, but nonetheless this experiment was unable to determine whether *tei* and *robo1* work together or in parallel because of the hypomorphic nature of knockdown experiments.

Q9. I am wondering why the authors keep changing the developmental stages of flies that were used for immunostaining, some with adult, and some with larvae or pupae. I do not have problem with using either developmental stage for certain experimental purpose, e.g., whether over-expression of FruBM would inhibit Tei expression in larvae or other stage, but the authors are at least should mentioned the logic between using flies with different developmental stages for different experiments.

A9. Regarding the description of the larval FruM overexpression experiment, we rephrased the passage as follows (lines 235–238): “We found a striking reduction in the intensity of anti-Tei staining upon FruM-overexpression in the third instar larval CNS (Fig. 6a-c vs. d-f), which otherwise does not express FruM and thus comparisons can be made without any interference from endogenous FruM expression”. Similarly, we rephrased the passage describing endogenous Tei expression as follows (lines 208–211): “We therefore confined the reporter expression to a fru-positive subset of cells by the intersection of *tei*-GAL4 and *fru*FLP and examined the CNS at the early pupal stage, when endogenous FruM expression commences to organize the sex-specific neural network (Fig. 5).”

Reviewer 3

Overall impression of the work

Q10. This manuscript contains a body of meticulously designed experiments that establishes Tei as the second gene target of FruBM. The authors presented a thought-provoking hypothesis that Tei may exert its function by forming a heteromeric complex with Robo, though the data presented may suggest otherwise—see my comments below. The hypothesis is interesting as the use of the immunoglobulin-like structural domain to generate functional variability is a recurrent theme observed in other fields.

I think the authors can develop this hypothesis further by discussing more about how Slit/Robo1 signaling works. For example, can Robo1 work on its own? What is the stoichiometry of a functional Robo1 receptor? Is there anything known about what the C-terminal domain of Robo1 does during Slit/Robo1 signaling? These questions are relevant to Tei as it does not have a C-terminal domain like Robo, according to Fig 1a. If Tei were to interact with Robo, how will this affect Slit/Robo1 signaling?

A10. We added the following sentence to discuss the possible role of the cytoplasmic C-terminus of Robo1, which is absent from Tei. “The lack of the cytoplasmic portion in Tei might suggest that Tei contributes to the ligand specificity whereas the specificity of intracellular signal transduction following heteromeric receptor activation is determined by Robo1 or another Tei partner that carries the C-terminal cytoplasmic domain” (lines 267–271).

Specific comments, with recommendations for addressing each comment

Q11. P7 line 101: What are the 7 genes (assuming CG17716 is one of the seven) that are downregulated? The cited Supplementary Table 1 does not seem to be the right table intended for this sentence.

A11. The 7 downregulated genes include CG17716 (*tei*), *Kr-h1*, *Tnks*, *robo3*, *Sema2a*, *Cip4* and *ds*, and the names of these genes are given in the revised text (lines 101–103).

Q12. P11 lines 167-168: You may want to emphasize that there are two *tei* RNAi lines that confirm this observation here, since you’re citing results from both lines.

A12. Three *tei* RNAi lines were used in this study to assure that the observed effects were truly due to *tei* knockdown, and this fact is now described in lines 140–144 of the original manuscript (corresponding to lines 143–146 of the revised manuscript). We also added text to the Methods subsection on fly lines in order to clarify that we made the two *tei* RNAi lines, *UAS-tei RNAi^{ex6-7}* and *UAS-teiRNA^{ex4}*, as well as *tei-GAL4* and *UAS-tei⁺* (lines 342–343).

Q13. P11-12 lines 170-182 and Figures 5d-g: Why did you use a different Gal4 line for the overexpression experiment? I assume *poxn-Gal4* is EY11779? Why not use *PPk23-Gal4* with *fru2/frusat*? I assume for practical reasons, but some explanation is needed.

A13. To clarify why we used different combinations of GAL4 and UAS constructs to overexpress *tei⁺* from one experiment to another, we modified the relevant passage as follows: “In this series of experiment, *tei⁺* was overexpressed with EY11779, a Gene-Search fly line with paired UAS sequences inserted near the *tei* locus on chromosome-2 (*UAS-tei[+]*), so as to make chromosome 3 available for other transgenes to combine. Also, *Poxn-GAL4*, a pan-chemosensory driver, replaced *ppk23-GAL4* in another set of experiment, because the chromosome-3 carrying both a *fru* mutant allele and *Poxn-GAL4* was already available with no need to generate a new recombinant chromosome-3” (lines 173–

179).

Q14. P12 lines 180-181: "... additional ... *tei+* in these neurons counteracted ... to increase midline crossing.." but you're actually looking for a decreased midline crossing in the cited Figure. I understand the meaning, but the sentence is a bit convoluted. Rewording or rephrasing may improve readability.

A14. We rephrased the relevant passage as "Importantly, additional expression of *tei+* in these neurons counteracted the effect of *robo1* RNAi, resulting in a trace of midline crossing (Fig. 4o-q)." (lines 188–190)

Q15. P12 lines 178-182, P16 lines 252-260 *FruBM* represses both *Robo1* and *Tei* activity. Both *Robo1* and *Tei* KD promote midline crossing. Results show that overexpression of *Tei* can counteract *Robo1* KD effect. Wouldn't this observation disprove the hypothesis stated in the abstract (p2 lines 29-30) that "*Tei* interacts with other IgG superfamily transmembrane proteins, including *Robo1*, to feminize the neurite patterns in females..."? If *Tei/Robo1* functions as a heterodimer, then KD of one gene should not be rescued by the OE of another? Can these receptors function as homodimers? *Tei* lacks a substantial C-terminal domain (Fig 1d). What does it mean when it interacts with other IgG transmembrane proteins with a larger C-terminal domain? Will it inhibit the intracellular effect completely or just attenuate it!?

A15. Both *robo*[+] and *tei*[+] prevent male-specific neurites from forming in females (they function as feminizers) and both are transcriptionally repressed by male-specific *FruBM* in males. In the revised manuscript, we extended our hypothesis and suggested the possibility that *Tei* without the C-terminal cytoplasmic domain may be important for the determination of ligand specificity at the extracellular side, whereas the C-terminal cytoplasmic domain of *Tei*-interacting IgG transmembrane proteins may specify intracellular signaling upon activation of the receptor heterodimer (lines 267–271).

Q16. P14 lines 210-219: The observation that the intensity of *Tei* signals in *FruM*-expressing cells is strong is very interesting. The authors offer two explanations: 1) *FruM* only represses *Tei* expression in a subset of mAL neurons; and 2) There's a temporal lag between *FruM* expression and *Tei* expression. Can't possibility 2 be tested by fine-tuning the developmental time points for the co-staining analysis in the experiments for Fig 6? If possibility 1 is true, then there must be an unidentified factor that stops *FruM* from repressing *Tei* expression in a subset of mAL neurons? Could this factor be *Fru/Bon/HP1a*?! For example, *FruM* is present but it's in complex with the unknown factor that prevents it from binding to the promoter of *Tei*?

A16. We expanded this part of the Results section based on our additional analysis of *FruM* and *Tei* expression in the adult brain by adding the following passage to the text (lines 228–230): "In the adult

brain, most of FruM-expressing mAL neurons (~20 cells) were also positive for tei-GAL4 presumably due to perdurance of GAL4 protein (Supplementary Fig. 6), an observation that appears to favor the second possibility”. The newly added Supplementary Fig. 6 shows a male-adult brain that was double-stained with the anti-Tei and anti-FruM antibodies in an immunostaining-enhancing solution (Immuno-shot; Cosmo Bio, IS-F-20); the fly used for staining carried an intersection triad: tei-GAL4, fru[FLP] and UAS>STOP>mCD8::GFP (highlighted by the box drawn with a white broken line is an mAL cluster that contained 18 Tei and FruM doubly positive neurons).

Q17. P14 lines 216-219: The Dsx experiment doesn't seem to belong in this section. Instead, the first paragraph of the next section (p15 lines 293-295) should go after line 216 on p14?

A17. We moved the entire description of the Dsx-single positive neurons so that it is included in the sentence reporting the effect of fru expression on tei expression (lines 231–233).

Q18. P20 Fly strains: explain somewhere what poxn-Gal4 is.

A18. We added a short explanation about poxn-GAL4 (lines 176–177; please also see A13 above).

Q19. P23 line 348: missing a bracket

A19. We changed the description as follows: “(residues 56–78 of Tei; GenBank accession number NP_523731).”

Q20. P24 line 377: “so that the circle was centered ...” how do you ensure this is done consistently from sample to sample?

A20. The following description was added to the Methods section to explain how to position circle-a (lines 421–424): “A few sensory neurons originating from prothoracic legs send ascending axons directly to the brain, which turn perpendicularly within the prothoracic ganglion. The midpoint of bilateral ascending fibers at their turning points was used to center the circle”.

Q21. P35 lines 534-537: More description of the arboplot would be nice. For example, is there any consistent landmark you use to draw the yellow dotted box for optical slicing for each sample? Does it need a special plugin in Image J? Maybe useful to write a method section to describe how this is done. It's a very nice quantification procedure that other researchers will find useful to implement.

A21. We added a supplementary Figure, the legend of which provides the following detailed explanation of the arboplot: “Supplementary Fig. 1. Arboplot analysis of mAL neurons. (a,b) The method to quantify the fluorescence distribution in single neural images. To scale the mAL neurites, we first determine the left and right ends of the paired antennal lobe in a given specimen (the contour of antennal lobes is highlighted by white broken lines), and the middle point of the two ends defines

the midline. We then draw a square, a side of which coincides with the half distance from the midline to a lateral edge of the antennal lobe on the neural image; the square is positioned so that its one vertical side superimposes on the midline while the upper side passes through the branching point at which the contralateral neurite bifurcates into the ascending and descending components (upper panel). The vertical (rostral) side of this square defines the y-axis (0-3) whereas the horizontal (lateral) side defines the x-axis (0-5 for a hemisphere) of the neural image subjected to fluorescence intensity measurements. We measure the fluorescent intensity of all pixels within the bottom one third (boxed with a yellow line) and the integrated intensity (0-1) for every x-axis point is plotted in the central to lateral direction as shown in the lower panel (arboplot). (c,d) Fluorescence intensity quantification with multiple neural images. Examples of the arboplots for ten different brains (c) and an average arboplot for the illustrated ten traces (d).” (lines 24–40)

Q22. P37 line 576: EY11779/Poxn-Gal4?

A22. In the revised manuscript, we explicitly stated that EY11779 is a Gene-Search fly line that functions as a UAS-tei[+] to overexpress tei[+], which is activated by poxn-GAL4, a pan-chemosensory GAL4 driver (lines 173–176). Please see also A13 above.

Q23. P38 Figure 6: How do you know the single cluster that expresses Tei in female is the same cluster that expresses Fru in male (Figure a-f), since their expression is supposedly mutually exclusive?

A23. This was almost the only cluster that expressed fru[FLP] as monitored with UAS>STOP>mCD8::GFP at this developmental stage in the entire CNS in both females and males, and therefore, this cluster was inferred to represent a homologous cluster in the two sexes.

Q24. P35 line 544, P40 line 611: “The box plot shows the median and 10th, 25th, 75th, 90th percentiles”? Please correct.

A24. The box-and-whisker plot shows the first quartile (25th percentile), median, third quartile (75th percentile) and minimum and maximum of each set of data. Thank you for pointing out this fundamental error. We corrected it in the revised manuscript.

REVIEWERS' COMMENTS:

Reviewer #1 (Remarks to the Author):

This is a revised manuscript by Sato et al., identifying a new *Fruitless* target gene, *teiresias*, that functions to feminize neuron structure in the *Drosophila* adult nervous system. The revisions address my concerns raised in the initial submission except one.

As a major point, it was asked that the authors provide specific optical sections for confocal images that could be influenced by the stacking of z-projections. As an example, I suggested SFig. 4. However, I was not initially clear and it isn't enough to add a line stating in the figure legend that a z-projection is shown. Rather it is critical to specify how many optical sections were used especially for Fig. 4 and Fig. 2b',d' where the midline crossing and outgrowth phenotype can be influenced by the number of optical sections. This needs to be done in the figure legends in the manuscript not in any supplementary information.

The authors did include copulation success as requested although *tsh-Gal80* was not included so the reader is left to wonder if this is a VNC or brain neuron phenotype.

Reviewer #2 (Remarks to the Author):

This revision fully addressed the two concerns I had previously.

Reviewer #3 (Remarks to the Author):

All my comments have been addressed adequately except for Q11. I don't quite understand how Seqs 1-16 in Supplementary Table 1 is related to the 7 downregulated genes described in the main text. Are these the search DNA sequences that were used to search for genes that have the Pal1 sequence motif in the genome? Maybe just take out the reference to this table on page 7 line 103 and also edit the title of this table?

Point-by-point replies to the reviewers' comments

Reviewer #1 (Remarks to the Author):

Q1. As a major point, it was asked that the authors provide specific optical sections for confocal images that could be influenced by the stacking of z-projections. As an example, I suggested SFig. 4. However, I was not initially clear and it isn't enough to add a line stating in the figure legend that a z-projection is shown. Rather it is critical to specify how many optical sections were used especially for Fig. 4 and Fig. 2b, where the midline crossing and outgrowth phenotype can be influenced by the number of optical sections. This needs to be done in the figure legends in the manuscript not in any supplementary information.

A1. In the re-revised manuscript (R2), we described, in respective figure legends, the number of confocal slices used in z-stack reconstitution for every image subjected to quantification of neural structures. It ranged between 29 and 100 slices, which, we believe, were sufficient to yield reliable estimates for the midline crossing score or fluorescent intensity score.

Q2. The authors did include copulation success as requested although *tsh-Gal80* was not included so the reader is left to wonder if this is a VNC or brain neuron phenotype.

A2. We attempted to repress GAL4 expression in the VNC with *tsh-GAL80*, but it turned out that *tsh-GAL80* on its own reduced the male copulation success rate down to ~75% (cf. 85% in *fru-GAL4/+* control flies) without *tei*⁺ overexpression, which was statistically indistinguishable from the value upon brain-restricted *tei*⁺ overexpression as shown in the figure pasted below. This made it difficult for us to judge which of the brain or VNC was the primary site of action of *tei* in modulating mating success.

NS: Not significant in the Fisher's exact probability test.

Reviewer #3 (Remarks to the Author):

Q3. All my comments have been addressed adequately except for Q11. I don't quite understand how Seqs 1-16 in Supplementary Table 1 is related to the 7 downregulated genes described in the main text. Are these DNA sequences that were used to search for genes that have the Pal1 sequence motif in the genome? Maybe just take out the reference to this table on page 7 line 103 and also edit the title of this table?

A3. We used Seqs 1-16 as query nucleotide sequences in our *in silico* searches for Pal1 homologous sequences in the genome. Seqs 1-16 were based on the FruBM binding motif (Pal1) identified in the *robo1* promoter (Ito et al., 2016), while allowed to carry two mismatches in the entire 16-bp palindrome sequence. Also, according to the reviewer's suggestion, we removed the reference to Supplementary Data 1 (former Supplementary Table 1) (line 96, page 7: former line 103) and modified the title of Supplementary Data 1 as "A list of query nucleotide sequences and the results of *in silico* similarity searches for genes carrying the Pal1 sequence motif".